# Structural basis for Ragulator functioning as a scaffold in membrane-anchoring of Rag GTPases and mTORC1

Tianlong Zhang[1], Rong Wang[1], Zhijing Wang[1,2], Xiangxiang Wang[3], Fang Wang[1] & Jianping Ding[1,2]

Amino acid-dependent activation of the mechanistic target of rapamycin complex 1 (mTORC1) is mediated by Rag GTPases, which are recruited to the lysosome by the Ragulator complex consisting of p18, MP1, p14, HBXIP and C7orf59; however, the molecular mechanism is elusive. Here, we report the crystal structure of Ragulator, in which p18 wraps around the MP1-p14 and C7orf59-HBXIP heterodimers and the interactions of p18 with MP1, C7orf59, and HBXIP are essential for the assembly of Ragulator. There are two binding sites for the Roadblock domains of Rag GTPases: helix α1 of p18 and the two helices side of MP1-p14. The interaction of Ragulator with Rag GTPases is required for their cellular co-localization and can be competitively inhibited by C17orf59. Collectively, our data indicate that Ragulator functions as a scaffold to recruit Rag GTPases to lysosomal membrane in mTORC1 signaling.

[1] State Key Laboratory of Molecular Biology, National Center for Protein Science Shanghai, Shanghai Science Research Center, CAS Center for Excellence in Molecular Cell Science, Institute of Biochemistry and Cell Biology, Shanghai Institutes for Biological Sciences, University of Chinese Academy of Sciences, Chinese Academy of Sciences, 320 Yue-Yang Road, Shanghai 200031, China. [2] School of Life Science and Technology, ShanghaiTech University, 393 Hua-Xia Zhong Road, Shanghai 201210, China. [3] School of Life Sciences, Shanghai University, 333 Nanchen Road, Shanghai 200444, China. Tianlong Zhang and Rong Wang contributed equally to this work. Correspondence and requests for materials should be addressed to J.D. (email: jpding@sibcb.ac.cn)

The mechanistic Target Of Rapamycin Complex 1 (mTORC1) coordinates cell growth, proliferation, and differentiation in response to environmental conditions of energy, nutrients, and extracellular growth factors[1]. Dysregulation of mTORC1 is often implicated in pathophysiological conditions such as tumorigenesis and diabetes[2]. Amino acids are particularly potent signaling molecules for activating mTORC1 in the anabolism and autophagy pathways, and small GTPases RagA-D play an essential role in rendering this signaling[3–5].

Rag GTPases are unique members of the Ras superfamily and function as obligate heterodimers such that RagA or RagB interacts with RagC or RagD through their C-terminal Roadblock domains[6, 7]. So far, several proteins and protein complexes, including the cytoplasmic leucine/arginine sensor proteins Sestrin2 and CASTOR1, and the downstream GATOR1, GATOR2, and KICSTOR complexes, have been reported to function as regulators of the Rag GTPases in response to amino acids[1]. In the supply of amino acids, the Rag GTPases are activated such that RagA or RagB is loaded with GTP and RagC or RagD is loaded with GDP, and thereafter promote the translocation of mTORC1 to the lysosomal membrane where the kinase activity of mTORC1 is stimulated by small GTPase Rheb[8, 9].

Unlike mTORC1, the Rag GTPases are localized to the cytoplasmic surface of the lysosomal membrane independent of amino acids[10]. Due to the lack of a transmembrane domain or a lipid-modified motif, the Rag GTPases are tethered to the lysosome by a lysosome-anchored protein complex called the Ragulator complex consisting of p18/LAMTOR1, p14/LAMTOR2, MP1/LAMTOR3, C7orf59/LAMTOR4, and HBXIP/LAMTOR5. All five components are necessary for the localization of Rag GTPases and mTORC1 to the lysosome and thus the activation of mTORC1[10, 11]. C17orf59, a component of the BORC complex, can decrease the lysosomal localization of Rag GTPases and impair the amino acid activation of mTORC1 via disruption of the Ragulator–Rag GTPases interaction[12]. Besides, the Ragulator complex may exert a guanine-nucleotide exchange factor (GEF) activity towards RagA and RagB and could be regulated by the lysosomal v-ATPase[11]. The potential lysosomal amino acid sensor SLC38A9 may interact with the Rag GTPases–Ragulator-v-ATPase complex and activate mTORC1 in response to arginine[13, 14]. The Ragulator-v-ATPase complex is also essential in LKB1-mediated AMPK activation under the regulation of glucose sensor aldolase, providing a switch between catabolism and anabolism[15, 16]. In the Ragulator complex, p18 is anchored to the lysosome through the N-terminal myristoylated and palmitoylated sites[17]. The other four components all comprise Roadblock domains, similar to the Rag GTPases. MP1 and p14 form a heterodimer as revealed by the crystal structure[11, 18, 19]. C7orf59 and HBXIP are suggested to form another heterodimer[11]. HBXIP alone exists in a monomer-dimer equilibrium in solution and assumes a homodimer in crystal structure[20]; however, the structural feature of C7orf59 remains unknown. The EGO complex in budding yeast, consisting of the Rag family GTPases Gtr1–Gtr2 and the Ego1–Ego2–Ego3 complex (EGO-TC), is also vital for amino acid-dependent activation of TORC1 and thus may be the counterparts of the Rag GTPases and the Ragulator complex[21, 22].

To explore the molecular mechanism by which the Ragulator complex recruits and activates the Rag GTPases and mTORC1, we determined the crystal structure of the Ragulator complex and identified two binding sites for the Rag GTPases, one of which is competitively inhibited by C17orf59. Structural comparison revealed a structural conservation between human Ragulator and yeast EGO-TC. Our structural and biological data together reveal the architecture of the Ragulator complex and the molecular basis for the functional role of Ragulator as a scaffold in the lysosomal membrane-anchoring of Rag GTPases and mTORC1.

## Results

**Structure of the Ragulator complex.** The N-terminal ~13 residues of p18 contain the myristoylation and palmitoylation sites which are essential for its association with the lysosomal membrane[10]. The lipidation sites truncated p18 (p18[14–161]) alone could not be obtained in soluble form as it expressed in *Escherichia coli* as inclusion bodies, but the N-terminally SUMO-tagged p18[14–161] could be co-expressed with MP1-p14 and further co-purified with C7orf59-HBXIP as a stable, monodisperse, heteropentameric complex as revealed by size exclusion chromatography – multi-angle static light scattering (SEC-MALS) analyses (Supplementary Fig. 1a). Various attempts to crystallize this complex were unsuccessful. Secondary structure prediction suggests that p18 comprises four α-helices (Supplementary Fig. 2a), and co-immunoprecipitation (co-IP) assays indicate that residues 1–48 of p18 are not required for its interactions with the other four components and residues 49–161 of p18 (p18[49–161]) comprising helices α1–α4 are necessary and sufficient for the interactions (Supplementary Fig. 2b). Thus, we prepared the Ragulator complex containing p18[49–161] which could be crystallized in space group $P2_12_12_1$. The structure of the Ragulator complex was solved to 2.9 Å resolution using the MR method, and there are two complex molecules in the asymmetric unit each with a 1:1:1:1:1 stoichiometry of the five components (Table 1 and Fig. 1a). The two molecules are structurally almost identical with a root-mean-square deviation (RMSD) of 0.5 Å for 482 Cα atoms, and molecule A that comprises more traced residues with better-defined electron density is used in the discussion hereafter.

In the Ragulator complex, p18 forms a helical structure over its entire traced length (residues 76–149) composed of helices α2-α4, and the N-terminal (residues 49–75) and C-terminal (residues 150–161) regions of the p18[49–161] construct are undefined in the electron density (Fig. 1a). MP1, p14, C7orf59, and HBXIP all assume Roadblock domain folds (Supplementary Fig. 3). Both MP1 and p14 assume a typical Roadblock domain fold (αββαββα), and the MP1-p14 heterodimer assumes a typical dimeric Roadblock domain fold composed of a ten-strand β-meander flanked by a four-helix bundle on one side (four helices side) and a two-helix bundle on the other (two helices side)[23], similar to that in the structure of MP1-p14 alone[18, 19] (Fig. 1b and Supplementary Fig. 4). However, both C7orf59 and HBXIP adopt an incomplete Roadblock fold (αββαβββ) missing the C-terminal α3 helix, and the C7orf59-HBXIP heterodimer form an incomplete Roadblock dimerization fold missing two α3 helices on the four helices side (Fig. 1b). Using co-expression, we also obtained a stable, monomeric C7orf59-HBXIP heterodimer (Supplementary Fig. 1b), and determined the crystal structure of C7orf59-HBXIP at 2.8 Å resolution (Table 1 and Supplementary Fig. 4). Compared with the MP1-p14 and C7orf59-HBXIP heterodimers alone, formation of the Ragulator complex does not cause notable conformational changes of the two heterodimers, except that the N-terminal helix α1 (residues 1–11) of C7orf59 is disordered in the heterodimer alone but is well defined in the complex owing to its interaction with p18 (Supplementary Fig. 4).

**Interactions among different components.** In the Ragulator complex, the five components have a very compact arrangement and the interaction interfaces bury a total solvent-accessible surface area of 12,608 Å[2]: the MP1-p14 and C7orf59-HBXIP heterodimers pack side by side via the four helices side of MP1-p14 and the two helices side of C7orf59-HBXIP, and p18 wraps around MP1, C7orf59, HBXIP, and p14 sequentially (Fig. 1a). Specifically, p18 makes extensive hydrophobic and hydrophilic interactions with the other four components (Figs. 1a, 2, and Supplementary Fig. 5). From the N-terminal end, helix α2

**Table 1 Summary of diffraction data collection and structure refinement statistics**

|  | HBXIP-C7orf59 | Ragulator (p18$^{49-161}$) | Ragulator (p18$^{76-145}$) |
|---|---|---|---|
| *Diffraction data* | | | |
| Wavelength (Å) | 1.0000 | 0.9793 | 0.9792 |
| Space group | P3$_2$21 | P2$_1$2$_1$2$_1$ | C222$_1$ |
| Cell parameters | | | |
| $a$ (Å) | 58.54 | 71.34 | 110.66 |
| $b$ (Å) | 58.54 | 89.62 | 117.85 |
| $c$ (Å) | 90.01 | 232.17 | 187.81 |
| Resolution (Å) | 50.0-2.80 | 50.0-2.90 | 50.0-2.65 |
|  | (2.90-2.80)$^a$ | (3.00-2.90) | (2.74-2.65) |
| Observed reflections | 29,043 | 125,559 | 136,964 |
| Unique reflections (I/σ(I) > 0) | 4,579 | 30,426 | 29,802 |
| Average redundancy | 6.3 (5.7) | 4.1 (4.2) | 4.6 (4.6) |
| Average I/σ(I) | 18.6 (2.8) | 16.7 (2.3) | 26.3 (2.1) |
| Completeness (%) | 97.1 (98.0) | 90.0 (88.4) | 98.1 (98.6) |
| $R_{merge}$ (%)$^b$ | 9.5 (47.7) | 10.5 (54.0) | 4.6 (50.9) |
| *Refinement and structure model* | | | |
| Reflections (Fo ≥ 0σ(Fo)) | | | |
| Working set | 4,163 | 27,099 | 25,086 |
| Test set | 207 | 1,410 | 1,637 |
| $R_{work}/R_{free}$ (%)$^c$ | 21.5/25.6 | 24.8/28.8 | 21.6/26.3 |
| No. of protein atoms | 1,205 | 7,411 | 6,750 |
| No. of water atoms | 10 | 12 | 58 |
| Average B factor (Å$^2$) | | | |
| All atoms | 50.7 | 62.4 | 61.5 |
| Protein | 50.8 | 62.4 | 61.5 |
| Water | 41.1 | 41.5 | 58.3 |
| RMS deviation | | | |
| Bond lengths (Å) | 0.008 | 0.007 | 0.006 |
| Bond angles (°) | 1.1 | 1.3 | 1.2 |
| Ramachandran plot (%) | | | |
| Favored | 96.2 | 96.0 | 96.7 |
| Allowed | 3.8 | 4.0 | 3.3 |
| Disallowed | 0.0 | 0.0 | 0.0 |

$^a$Numbers in parentheses refer to the highest resolution shell
$^b$R$_{merge}$ = $\sum_{hkl}\sum_i |I_i(hkl)_i - \langle I(hkl) \rangle| / \sum_{hkl}\sum_i I_i(hkl)$
$^c$R factor = $||F_o| - |F_c|| / |F_o|$

**Functional validation of the interactions**. To validate the biological relevance of the Ragulator structure, we first performed co-IP assays to investigate whether truncations or mutations of key residues on p18 would affect formation of the Ragulator complex. First, we examined the interface of p18 with MP1. Truncation of the MP1-binding region on p18 (p18$^{108-161}$) completely abolishes the interactions with MP1 and p14, and weakens the interactions with C7orf59 and HBXIP (Fig. 3a). Nonetheless, several single or multiple mutations of the residues on p18 have no significant effects on the interactions probably because there are many residues involved in the interactions (Supplementary Fig. 7a). Second, we examined the interface of p18 with the C7orf59-HBXIP heterodimer. Mutation L119R or V132D on p18 in the C7orf59- or HBXIP-binding region significantly impairs the interactions with the other four components (Fig. 3b). Finally, we examined the interface of p18 with p14. As p18 has very few interactions with p14, it is not surprising to observe that removal of the p14-binding region (p18$^{76-145}$) has no significant effect on the formation of the Ragulator complex (Fig. 3a). However, further shortening of the C-terminal region of p18 covering the HBXIP- or C7orf59-HBXIP-binding regions (p18$^{14-128}$ or p18$^{14-107}$) completely disrupts the formation of the Ragulator complex (Fig. 3a). The interface between MP1-p14 and C7orf59-HBXIP is primarily mediated by MP1, p14, and HBXIP (Supplementary Fig. 6). Mutation L102D on MP1 at the interface moderately compromises, and M103D on p14 and L113R on HBXIP substantially impair the formation of the Ragulator complex (Supplementary Fig. 7b).

To further verify the interactions of p18 with the other components in vivo, we constructed a p18 variant (p18$^{mito}$) in which the N-terminal lipidation region (residues 1–13) was removed but the mitochondrial transmembrane region of OMP25 was attached to the C-terminus, and performed immunofluorescence assays to examine the co-localization of the other components with p18$^{mito}$ to the mitochondria based on the method described previously[10]. As expected, RFP-p18$^{mito}$ localizes to the mitochondria, and MP1-p14 and C7orf59-HBXIP co-localize well with p18$^{mito}$ but not with the RFP-Mito control, suggesting that p18$^{mito}$ can recruit the other components to the mitochondria (Fig. 3c and Supplementary Fig. 8). Consistent with our co-IP assays, the p18 mutants with truncations or mutations of the binding region(s) for MP1, C7orf59, and HBXIP but not p14 cannot recruit MP1-p14 and C7orf59-HBXIP to the mitochondria (Fig. 3c).

These results indicate that the region consisting of residues 76–145 of p18 (from α2 to α4) is the minimum fragment to interact with the other components and form a stable pentameric Ragulator complex. To confirm this result, we constructed a p18$^{76-145}$ variant and obtained a stable Ragulator complex, and then solved its structure at 2.65 Å by the MR method (Table 1 and Supplementary Fig. 9a). Despite lacking the interaction between p18 and p14, this structure is essentially identical to the complex containing p18$^{49-161}$ (Supplementary Fig. 9b). Taken the structural and functional data together, we conclude that the interactions of p18 with MP1, C7orf59 and HBXIP are critical for the assembly of the Ragulator complex whereas the interaction with p14 is not. The interactions between MP1-p14 and C7orf59-HBXIP are largely dictated by p18, but their interactions also play an important role in the stabilization of the Ragulator complex.

(residues 77–97) of p18 packs along the bottom side of MP1 and interacts with helices α1 and α3, strand β2, and the β1–β2 loop of MP1. The following long loop (residues 98–114) of p18 makes a sharp turn (~90°) and stacks on helix α1 of MP1 and then protrudes into a cleft formed by strand β4 and the α2-β3 loop of C7orf59. Afterwards, helix α3 (residues 115–120) of p18 lies along the hydrophobic surface of the β-sheet of C7orf59 and interacts with helix α1 of C7orf59 to form a two-helix bundle. Similarly, helix α4 (residues 126–145) of p18 lies on the hydrophobic surface of the β-sheet of HBXIP and forms a two-helix bundle with helix α1 of HBXIP. Finally, the C-terminal loop (residues 146–149) of p18 lies on the surface of helices α1 and α3 of p14, and the C-terminal end (Asp149) is impeded by the hydrophobic side chains of Val10 and Tyr114 of p14, leading to the rest of the C-terminal region (residues 150–161) disordered in the electron density.

Besides p18, the Ragulator complex is also stabilized by extensive hydrophobic and hydrophilic interactions between the two heterodimers via the four helices side of MP1-p14 and the two helices side of C7orf59-HBXIP (Fig. 1a and Supplementary Fig. 6). Briefly, the β1-β2 loop of HBXIP protrudes into a cleft formed by the two α3 helices of MP1 and p14; helix α3 of MP1 interacts with helix α2 of HBXIP; and helices α1 and α3 of p14 interact with helix α1 and strand β2 of HBXIP.

**Ragulator interacts with Rag GTPases via two binding sites**. The Ragulator complex is responsible for the recruitment of Rag GTPases to the lysosome in mTORC1 signaling[11, 17]. We next analyzed the interactions between the Ragulator complex and the Rag GTPases using co-IP and immunofluorescence assays.

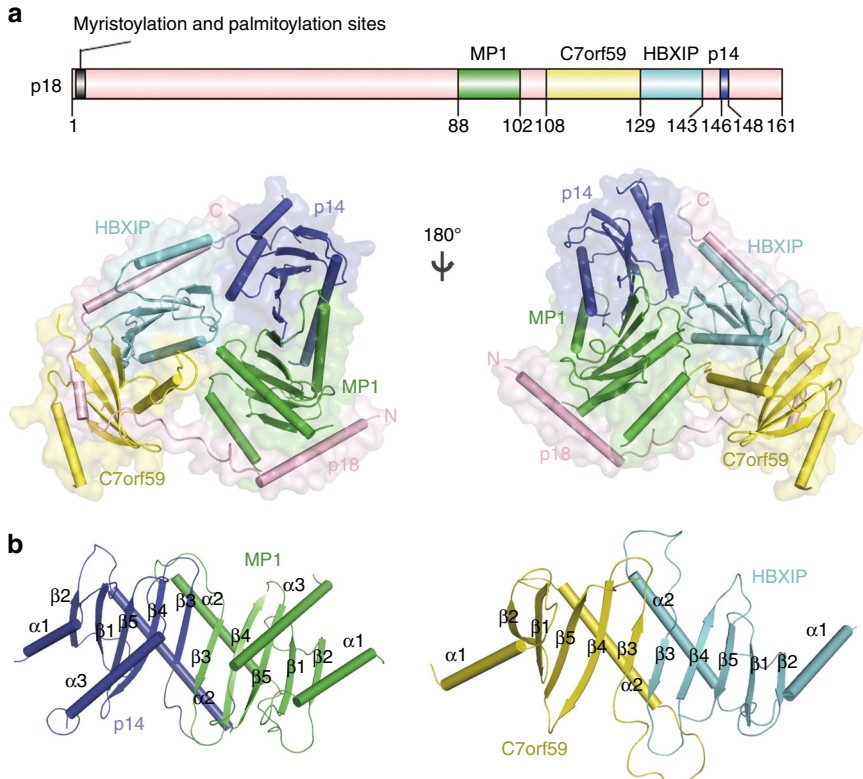

**Fig. 1** Crystal structure of the Ragulator complex. **a** Overall structure of the Ragulator complex in two different views. The Ragulator complex is shown as cylindrical cartoon model in a transparent envelope surface with p18 colored in pink, MP1 in green, p14 in blue, HBXIP in cyan, and C7orf59 in yellow, respectively. A schematic diagram of p18 is shown on the top with the interacting regions with MP1, C7orf59, HBXIP, and p14 are indicated. **b** Cylindrical cartoon presentations of the MP1-p14 and C7orf59-HBXIP heterodimers in the Ragulator complex. The secondary structure elements are labeled

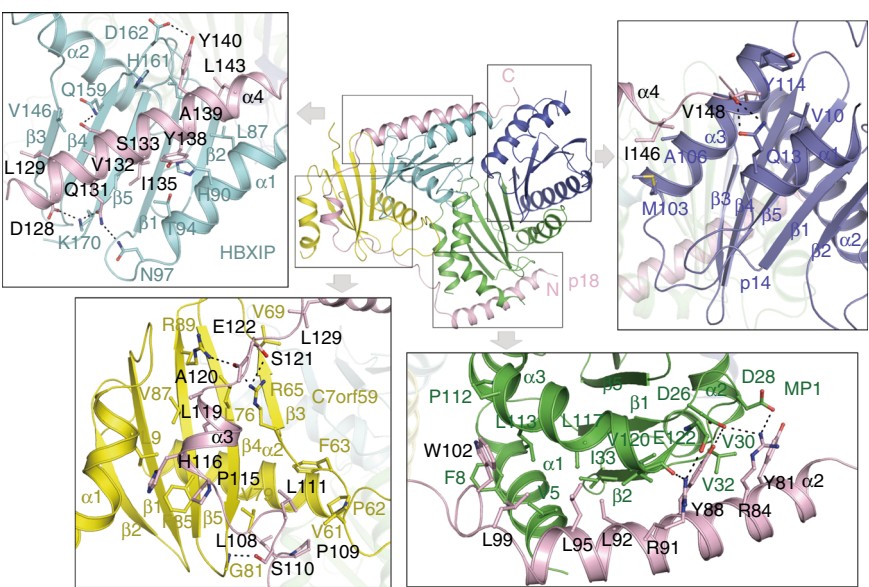

**Fig. 2** Detailed interactions of p18 with MP1, C7orf59, HBXIP, and p14. The proteins are colored as in Fig. 1a. The residues involved in the interactions are shown in ball-and-stick models. The hydrophilic interactions are indicated with dashed lines

Consistent with the previous results[10], both the dominant positive RagA$^{GTP}$–RagC$^{GDP}$ and dominant negative RagA$^{GDP}$–RagC$^{GTP}$ complexes exhibit comparable abilities to interact with the Ragulator complex and co-localize with the RFP-p18$^{mito}$-labeled Ragulator to the mitochondria, suggesting that the Rag GTPases interact with the Ragulator in a nucleotide-independent manner

(Supplementary Fig. 10). Each Rag GTPase consists of an N-terminal GTPase domain and a C-terminal Roadblock domain. The previous co-IP assays showed that dimerization of the Roadblock domains of Rag GTPases is necessary and sufficient for the interaction with p18[7]. Our co-IP and co-localization assays further show that the Ragulator binds to the Roadblock

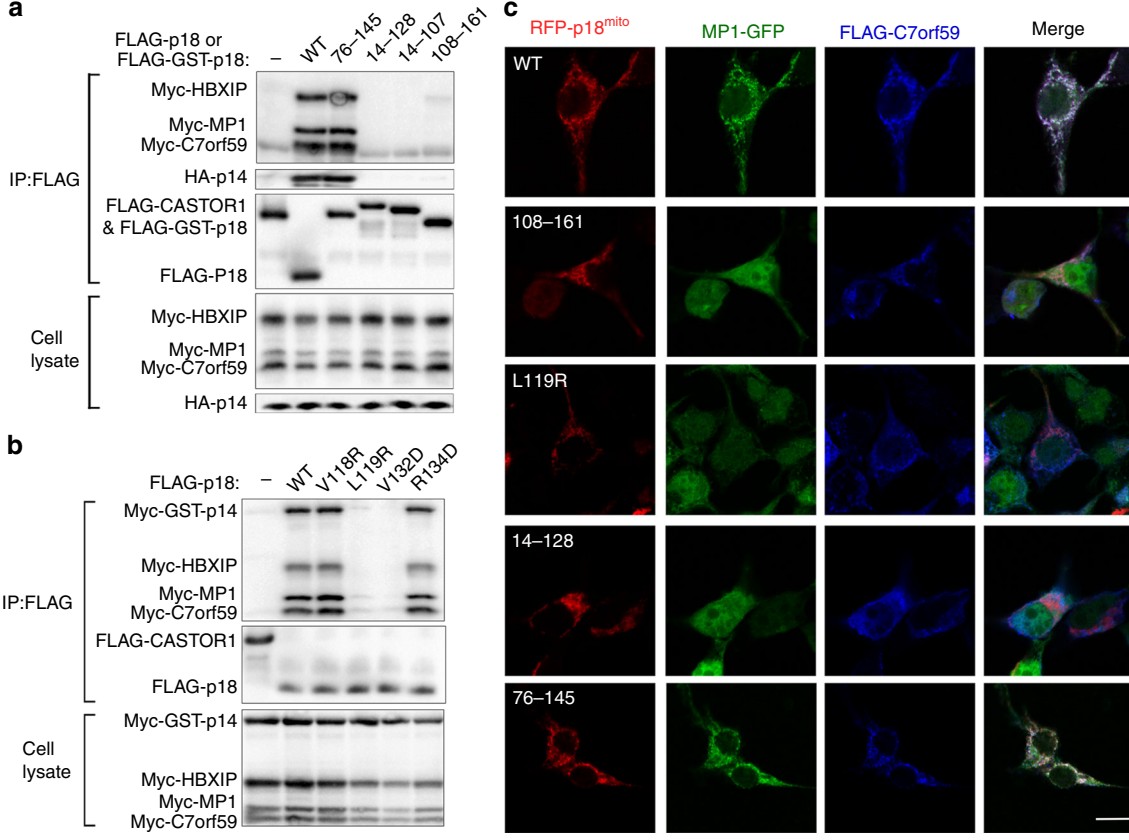

**Fig. 3** Validation of the interactions among different Ragulator components. **a** Co-IP assays to validate the interactions of p18 with the other four components. FLAG-tagged CASTOR1, the cytoplasmic arginine sensor that has no direct interaction with the Ragulator or the Rag GTPases, was used as a negative control. The p18[14-161] variant that lacks the N-terminal lipid-modification region was denoted as the wild-type (WT) protein. WT p18 was fused with a FLAG tag and the p18 variants were fused with a FLAG-GST tag to improve expression. **b** Co-IP assays to validate the interactions of p18 with C7orf59 and HBXIP. Mutation L119R or V132D on p18 significantly impairs the interactions with the other four components; as negative controls, mutations V118R and R134D exhibit no notable effect on the interactions because the two residues have no direct interactions with C7orf59-HBXIP. **c** Cellular co-localization of WT and mutant p18 with the other four components. The interacting regions of p18 for MP1, HBXIP, and C7orf59 are essential for the assembly of the Ragulator complex, but the region for p14 is not. Scale bar represents 10 μm

domains instead of the GTPase domains of Rag GTPases (Supplementary Fig. 11).

It was previously suggested that p18 is the principal Rag GTPase-binding component in the Ragulator[10]. Thus, we first tried to map the Rag GTPase-binding region on p18. With various p18 truncations, we demonstrate that helix α1 (residues 49–62) is essential for the binding of Rag GTPases (Fig. 4a). As this helix is disordered in the solved structures, we speculate that it might have high flexibility without Rag GTPases binding. Sequence analysis shows that helix α1 contains two highly conserved amino acid patches (Supplementary Fig. 2a). Our mutagenesis data show that the p18 mutant (M2) containing mutations I57D and L58D on helix α1 can form a stable Ragulator complex with the other four components, but cannot interact with and recruit the Rag GTPases to the mitochondria (Figs. 4c, d and Supplementary Table 1).

Intriguingly, the p18[14–75] variant comprising the Rag GTPase-binding region fails to interact with the other four components and the Rag GTPases (Fig. 4a), and additionally, overexpression of the RFP-p18[mito] alone cannot recruit the Rag GTPases to the mitochondria (Supplementary Fig. 12a). Furthermore, the previous biological data showed that lacking any of MP1, p14, C7orf59, and HBXIP, the Rag GTPases could not localize to the lysosome[10, 11]. These results led us to speculate that besides p18, MP1-p14 or/and C7orf59-HBXIP might also have direct interaction with the Rag GTPases. Previous structural and functional

studies showed that the Roadblock domain containing MglB homodimer possesses GAP activity for small GTPase MglA, and the longin domain containing Mon1-Ccz1 heterodimer and TRAPP complex are GEFs for small GTPases Ypt7 and Ypt1, respectively[24–26]. Longin domains assume the structural fold of ββαββαα and share similar structural features as the Roadblock domains (αββαββα)[27, 28]. In the above three complexes, the Roadblock or longin domains interact with their substrates via the two helices sides (Supplementary Fig. 13a). In the Ragulator complex, the two helices side of C7orf59-HBXIP is shielded by MP1-p14, whereas that of MP1-p14 is exposed to the solvent and is located closely to the N-terminal region of p18, and thus might be involved in the interaction with Rag GTPases (Fig. 1b). To examine this hypothesis, we mutated several groups of highly conserved residues on the two helices side of MP1 and p14 and performed co-IP and co-localization assays to identify the Rag GTPase-binding region(s) (Fig. 4b and Supplementary Table 1). The results show that these mutations do not affect formation of the Ragulator complex; however, two neighboring mutations M5 on MP1 and M7 on p14 abolish the interactions of the Ragulator with the Rag GTPases and impair the cellular co-localization of the Rag GTPases, indicating that the α2 helices of MP1 and p14 are also essential for the interaction and recruitment of the Rag GTPases (Figs. 4c, d). Nevertheless, in the absence of p18, the forced mitochondria localized constructs of MP1-p14 (MP1[mito]) and C7orf59-HBXIP (C7orf59[mito]) are deficient in the

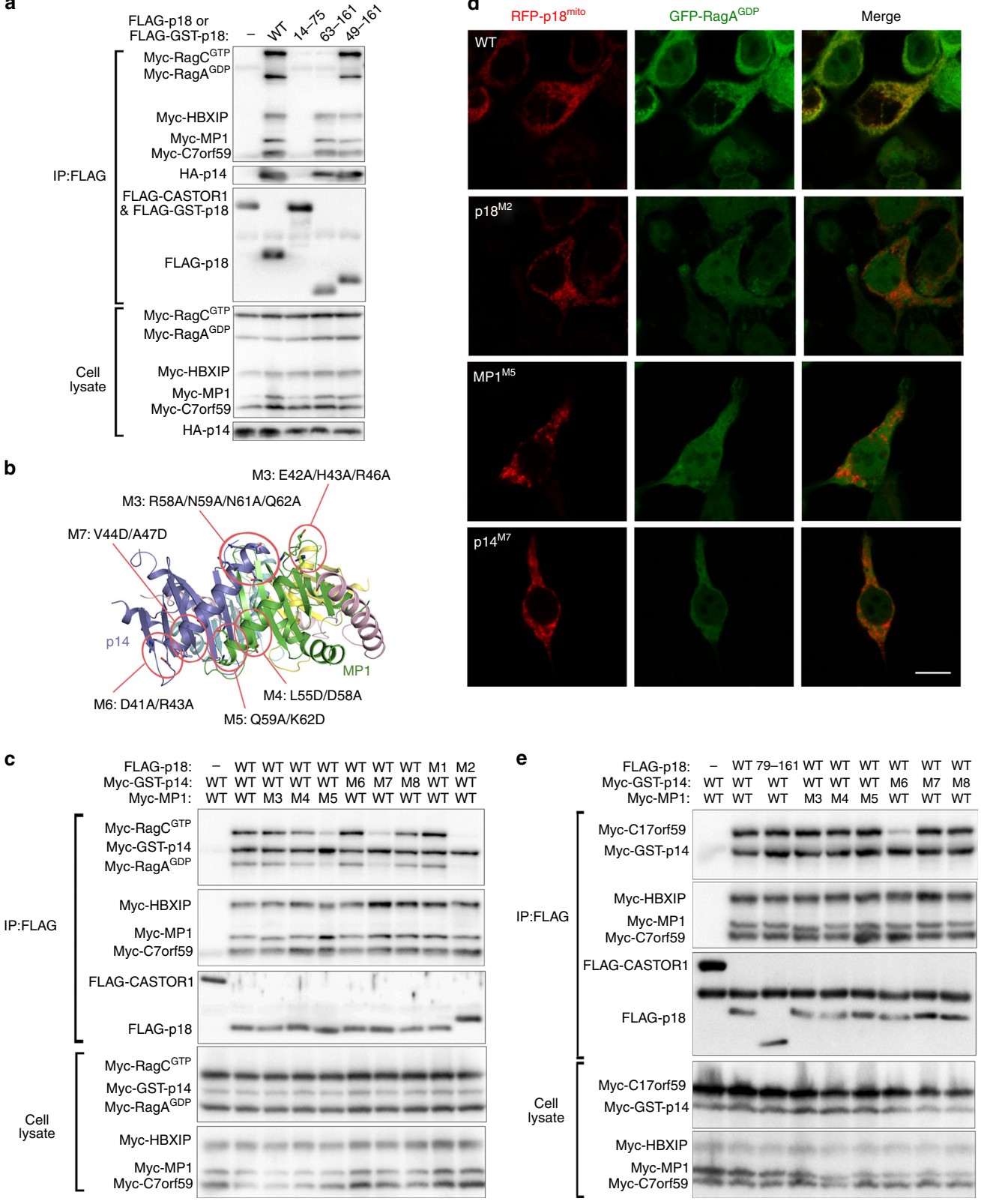

**Fig. 4** Mapping of the Rag GTPase-interacting sites on the Ragulator complex. **a** Co-IP assays of WT and p18 variants with the other Ragulator components and RagA$^{GDP}$–RagC$^{GTP}$. **b** A ribbon diagram of the MP1-p14 heterodimer showing the locations of mutations used in the mapping of the Rag GTPase-interacting site on the MP1-p14 heterodimer. **c** Co-IP assays of WT and mutant p18, MP1 or p14 with the other Ragulator components and RagA$^{GDP}$–RagC$^{GTP}$. **d** Localization of RagA$^{GDP}$–RagC$^{GTP}$ in cells over-expressing the Ragulator complex with WT and mutant p18, MP1 or p14. Scale bar represents 10 μm. **e** Co-IP assays of WT and mutant p18, MP1, or p14 with the other Ragulator components and C17orf59

recruitment of the Rag GTPases (Supplementary Fig. 12b, c). These results together demonstrate that both the N-terminal region of p18 and the two helices side of MP1-p14 are required for the interaction and recruitment of the Rag GTPases.

**C17orf59 competitively interacts with MP1-p14.** Recently, it was shown that C17orf59, a component of the BORC complex, is a negative regulator of the mTORC1 signaling upon amino acid activation via disruption of the Ragulator–Rag GTPases interaction[12]. To investigate the underlying mechanism, we examined the interaction between the Ragulator and C17orf59, and found that the p14 mutant M6 containing mutations D41A/R43A on helix α2 impairs the Ragulator-C17orf59 interaction, whereas the N-terminal truncation of p18 (residues 76–161) is capable of binding C17orf59, suggesting that helix α2 of p14 is the potential binding site for C17orf59 (Fig. 4e). As the C17orf59-binding and Rag GTPase-binding sites on p14 are located adjacently to each other (Fig. 4b), it is very likely that C17orf59 binds competitively to helix α2 of p14 and hence impedes the binding of Rag GTPases to the Ragulator complex.

## Discussion

The Ragulator complex is responsible for the recruitment of the Rag GTPases and mTORC1 to the lysosome and thus plays an important role in mTORC1 signaling. In this work, we determined the crystal structure of the Ragulator complex, which together with the functional data reveals the architecture of the Ragulator complex. We found that the lysosome-anchored p18

functions as a scaffold to recruit the MP1-p14 and C7orf59-HBXIP heterodimers to the lysosome and then to form the Ragulator complex. Among the extensive interactions, the interfaces of p18 with MP1, C7orf59, and HBXIP are critical for the assembly of the Ragulator complex but that with p14 is not. The interactions between MP1-p14 and C7orf59-HBXIP are largely dictated by p18, but their interactions also play an important role in the stabilization of the Ragulator complex. Disruption of the interaction of p18 with one heterodimer also disrupts its interaction with the other, suggesting that p18 interacts with MP1-p14 and C7orf59-HBXIP in a cooperative manner. We also identified two binding sites in the Ragulator, namely the α1 helix of p18 and the α2 helices of MP1-p14, which interact directly with the C-terminal dimeric Roadblock domains but not the N-terminal GTPase domains of Rag GTPases, and found that the interactions of the Ragulator with the Rag GTPases are required for their cellular co-localization and are essential in the recruitment of the Rag GTPases to the lysosome. These results provide new insights into the functional role of the Ragulator complex in mTORC1 signaling.

A previous study suggested that the Ragulator complex, extracted from HEK293T cells, might possess a GEF activity towards RagA and RagB[11]. Our co-IP and co-localization data show that the Ragulator complex interacts with the Roadblock domains instead of the GTPase domains of Rag GTPases (Supplementary Fig. 11). In addition, our biochemical data also show that the recombinant Ragulator complex exhibits no measurable GEF activity towards RagA–RagC (Supplementary Fig. 14). Furthermore, consistent with the previous results[11], our biochemical

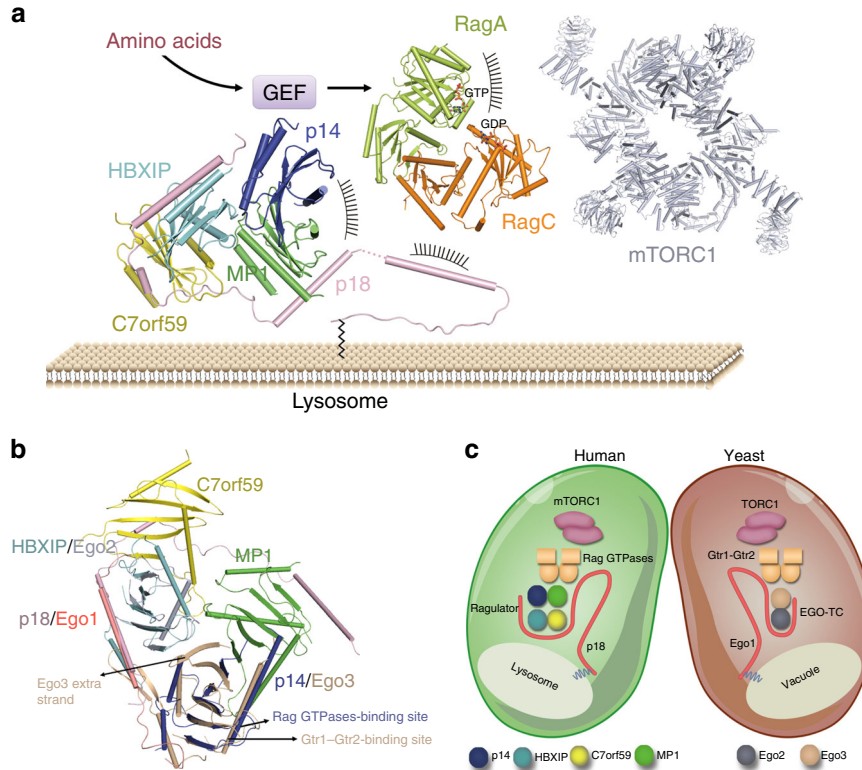

**Fig. 5** The Ragulator functions as a scaffold to recruit the Rag GTPases to the lysosomal membrane in mTORC1 signaling. **a** Working model for the functional role of the Ragulator complex in targeting the Rag GTPases to the lysosome. The model is constructed based on the structures of the Ragulator complex, the modeled Rag GTPases from yeast Gtr1–Gtr2 complex (PDB code 4ARZ), and the cryo-EM structure of mTORC1 (PDB code 5H64). **b** Superposition of the structures of the Ragulator and EGO-TC (PDB code 4XPM) complexes. The Ragulator components are colored as in Fig. 1a, and Ego1, Ego2, and Ego3 of the EGO-TC are colored in red, gray, and eggshell, respectively. **c** The Ragulator and EGO-TC complexes share similar structural and functional features in mTORC1/TORC1 signaling. Ego1, Ego2, and Ego3 of the EGO-TC appear to be the counterparts of p18, HBXIP, and p14 of the Ragulator

data also show that the Ragulator complex can interact with both RagA$^{GTP}$–RagC$^{GDP}$ and RagA$^{GDP}$–RagC$^{GTP}$ although it pulls down more RagA$^{GDP}$ than RagA$^{GTP}$ (Supplementary Fig. 10), which is different from other GEFs that bind preferentially to GDP-bound small GTPases. These results together indicate that the Ragulator complex alone does not function as a GEF for the Rag GTPases. If the Ragulator complex was a bond fide GEF, it might require additional component(s) to exert the GEF activity for the Rag GTPases or employs a novel mechanism. The Ragulator complex might also act as a scaffold for an unknown GEF which can interact preferentially to the inactive Rag GTPases and exert a GEF activity.

Based on our structural and functional data, we can propose a working model for the functional role of the Ragulator as a scaffold in the recruitment of Rag GTPases and mTORC1 to the lysosomal membrane (Fig. 5a). p18 is tethered to the lysosome through the N-terminal lipid-modifications, and then anchors the MP1-p14 and C7orf59-HBXIP heterodimers with its C-terminal regions to assemble the Ragulator complex. Subsequently, the Ragulator can recruit the Rag GTPases to the lysosome via its interactions with the C-terminal Roadblock domains of Rag GTPases. Upon the activation of amino acids, the N-terminal GTPase domains of the Rag GTPases are activated by an unknown GEF, and then the Rag GTPases in the active form can interact with mTORC1, leading to the activation of the mTORC1 signaling.

Previously, we have reported the crystal structure of yeast EGO-TC and revealed the structural conservation of the components between the EGO-TC and the Ragulator[22]. In the EGO-TC structure, Ego2 and Ego3 form a heterodimer with one side flanked by the C-terminal region of Ego1. Similar to p18, Ego1 is anchored to the vacuolar membrane through an N-terminal lipid modification. The other components of Ragulator and EGO-TC all contain structurally conserved Roadblock domain (Supplementary Fig. 3). Compared with the Ragulator proteins, Ego3 appears to be the counterpart of MP1 and p14 but has insertions of an extra β-hairpin and β-strand, assuming the αββββαββββα-fold, and Ego2 appears to be the counterpart of C7orf59 and HBXIP but lacks the N-terminal α-helix, assuming the ββαβββ-fold. Intriguingly, the EGO-TC can be superimposed onto the Ragulator very well with an RMSD of 2.1 Å for 196 Cα atoms: Ego1 overlaps with p18, while Ego2 and Ego3 occupy the spatial positions of HBXIP and p14, respectively, suggesting that the EGO-TC and the Ragulator are structurally conserved (Fig. 5b). Moreover, we previously identified an Ego3 mutant containing mutations N67A/N68A/K70A/M71A on helix α2 which fails to interact with Gtr1–Gtr2, the yeast homologs of Rag GTPases. This region of Ego3 is equivalent to the region of p14 interacting with the Rag GTPases, suggesting that the two complexes might interact with Rag/Gtr GTPases in a similar manner. However, the EGO-TC lacks two components corresponding to MP1 and C7orf59 of the Ragulator, and thus lacks the Rag GTPase-binding site on MP1. It is plausible that there are still other components of the EGO complex unidentified yet. However, we could obtain a stable Ego1–Ego2–Ego3–Gtr1–Gtr2 complex in vitro (unpublished results), indicating that the EGO-TC is sufficient to interact with Gtr1–Gtr2 and thus is the counterpart of the Ragulator. As the extra β-strand of Ego3 is located very closely to the Gtr1–Gtr2-binding region on helix α2, it is possible that this β-strand may partially mimic the function of MP1 to interact with Gtr1–Gtr2 (Fig. 5b). Taken together, these results suggest that the Ragulator and EGO-TC are structurally conserved and might exert similar function(s) via similar mechanism in the mTORC1/TORC1 signaling (Fig. 5c).

mTORC1 is a key downstream effector of many oncogenic pathways and has been implicated in the progression of cancers and diabetes[1]. Rag GTPases, especially RagD, promote tumor growth by over-activating the mTORC1 signaling[29]. As a key activator of the RagGTPases and mTORC1, the Ragulator is also involved in tumorigenesis, and mutations in all five components of the Ragulator have been identified in COSMIC database based on cancer genome sequencing[1]. For examples, MP1 is highly expressed in the ER-positive breast cancer cells and is essential for the survival of related cancer cells[30], and HBXIP promotes the growth and migration of breast cancer cells[31]. Thus, the Ragulator could be a potential target for drug design. In particular, the identified Rag GTPase-binding sites on the Ragulator might be good targets for small molecules or peptides as inhibitors for its interaction with and thus the recruitment of Rag GTPases and the activation of mTORC1.

## Methods

**Plasmid construction and protein purification.** The DNA fragments encoding the Ragulator components and Rag GTPases were amplified by PCR from the cDNA library of human cells. p18 variants containing residues 14–161, 49–161, and 76–145 were cloned into a modified pET-28a vector (Novagen) with an N-terminal His$_6$-SUMO tag. MP1 and p14 were cloned into the cloning sites I and II of pET-Duet vector (Novagen), respectively. C7orf59 was cloned into pET-28a with or without a C-terminal His$_6$-tag, and the C-terminal Roadblock domain of HBXIP$^{83-173}$ was constructed into pET-Duet vector. p18, MP1, and p14 were co-expressed in *E. coli* BL21 (DE3) cells (Tiangen) and the transformed cells were grown at 37 °C in LB medium until OD$_{600}$ reached 0.8 and then induced with 0.3 mM IPTG for 20 h at 16 °C. C7orf59 and HBXIP were co-expressed with the same method. The cells were harvested by centrifugation, resuspended in a lysis buffer (30 mM Tris-HCl, pH 7.5, and 200 mM NaCl) and then lysed by sonication. The C7orf59-HBXIP complex was purified by affinity chromatography using a Ni-NTA column (Qiagen) and further by gel filtration using a Superdex 75 10/300 (preparative grade) column (GE Healthcare). To obtain the Ragulator complex, cells of over-expressing p18-MP1-p14 and C7orf59-HBXIP were mixed together and then lysed by sonication in the lysis buffer. The complex was purified by affinity chromatography. After that, the N-terminal His$_6$-SUMO tag of p18 in the Ragulator complex was removed by a ubiquitin-like protease (ULP1) and the complex was further purified by gel filtration using a Superdex 200 16/60 column (preparative grade) (GE Healthcare) pre-equilibrated with a storage buffer (10 mM HEPES, pH 7.5, 100 mM NaCl, and 1 mM DTT). RagC and RagA were cloned into the cloning sites I and II of pET-Duet vector, respectively, with an N-terminal His$_6$-tag fused to RagC, and were co-expressed and purified with the same methods described above. Primers used to construct plasmids and generate mutants are listed in Supplementary Table 2.

**SEC-MALS analysis.** Purity, molecular mass and size distribution of the purified proteins were analyzed using an analytical light scattering instrument (SEC-MALS) equipped with an Agilent 1260 Infinity Isocratic Liquid Chromatography System, a Wyatt Dawn Heleos II Multi-Angle Light Scattering Detector (Wyatt Technology), and a Wyatt Optilab T-rEX Refractive Index Detector (Wyatt Technology). Analytical size exclusion chromatography was performed at room temperature using a Superdex 200 10/300 GL column (GE Healthcare) equilibrated with a buffer containing 10 mM HEPES (pH 7.5) and 100 mM NaCl. 50 μl protein sample at concentration of about 1.5 mg ml$^{-1}$ was injected with a flow rate of 0.5 ml min$^{-1}$. The column effluent was monitored in-line with three detectors that simultaneously monitor the UV absorption, light scattering, and refractive index. The measurements were analyzed using the ASTRA 6.1 software (Wyatt Technology) to determine the molecular masses of the proteins.

**Crystallization and structure determination.** Crystallization of the C7orf59-HBXIP heterodimer and the Ragulator complex was performed using the hanging drop vapor diffusion method by mixing 1.5 μl protein solution (about 15 mg ml$^{-1}$) and 1.5 μl reservoir solution at 16 °C. Crystals of the C7orf59-HBXIP heterodimer were grown from drops consisting of a reservoir solution of 0.16 M ammonium sulfate, 0.08 M sodium acetate trihydrate (pH 4.6), and 20% (w/v) polyethylene glycol (PEG) 4000. Crystals of the Ragulator complex containing p18$^{49-161}$ were obtained from drops containing a reservoir solution of 0.2 M ammonium sulfate, 0.1 M Bis-Tris (pH 5.5), and 25% (w/v) PEG 3350. Crystals of the Ragulator complex containing p18$^{76-145}$ were obtained from drops containing 0.2 M ammonium sulfate and 20% (w/v) PEG 3350. Diffraction data were collected at −175 °C at BL17U1 of Shanghai Synchrotron Radiation Facility and BL19U1 of National Facility for Protein Science in Shanghai, China, and were processed, integrated, and scaled together with HKL2000[32]. Statistics of the diffraction data are summarized in Table 1.

All the structures were solved by the molecular replacement (MR) method implemented in Phenix[33]. Structure refinement was carried out using Phenix and Refmac5[33, 34]. Model building was performed manually using Coot[35] with high-

quality electron density (Supplementary Fig. 15). Structural analysis was carried out using programs in CCP4[36]. Figures were generated using Pymol (http://www.pymol.org). Statistics of the structure refinement and quality of the final structure models are summarized in Table 1.

**Co-immunoprecipitation assay**. 3,500,000 human embryonic kidney cells (HEK293T) cells were plated in 10-cm dishes and cultured in DMEM (Hyclone) supplemented with 10% fetal bovine serum (Biochrom). All cDNAs were cloned into the pcDNA 3 vector. The cells were transfected separately with the following plasmids using Lipofectamine 2000 (Invitrogen). For analysis of the interactions among the Ragulator components, the cells were transfected with the following plasmids: 5 μg FLAG-CASTOR1 or FLAG-p18[14–161] or FLAG-p18[14–161] mutants or FLAG-GST-p18 truncations; 15 μg Myc-MP1 or Myc-MP1 mutants; 5.5 μg HA-p14 or Myc-GST-p14 or Myc-GST-p14 mutants; 3.3 μg Myc-HBXIP or Myc-HBXIP mutants; 2 μg Myc-C7orf59. For analysis of the interactions between the Ragulator and Rag GTPases, the cells were transfected with the following plasmids: 3 μg Myc-RagA T21N (denoted as RagA[GDP]) or Myc-RagA Q66L (RagA[GTP]); 3 μg Myc-RagC S75N (RagC[GDP]) or Myc-RagC Q120L (RagC[GTP]); 3.65 μg FLAG-CASTOR1 or FLAG-p18[14–161] or FLAG-p18[14–161] mutants; 12 μg Myc-MP1 or Myc-MP1 mutants; 4 μg Myc-GST-p14 or Myc-GST-p14 mutants or HA-p14; 3.3 μg Myc-HBXIP; 2 μg Myc-C7orf59. For analysis of the interactions between the Ragulator and the Roadblock domain (Rd) or GTPase domain (Gd) of RagA or RagC, the cells were transfected with the following plasmids: 5 μg Myc-RagA Gd[GDP] or 3 μg Myc-RagA Gd[GTP] or 3 μg Myc-RagA Rd; 1.2 μg Myc-RagC Gd[GTP] or 1.2 μg Myc-RagC Gd[GDP] or 3 μg Myc-RagC Rd; 3.65 μg FLAG-p18[14–161]; 9.6 μg HA-MP1; 4 μg Myc-GST-p14; 1 μg HA-HBXIP; 4.8 μg HA-C7orf59. For analysis of the interactions between the Ragulator and C17orf59, the cells were transfected with the following plasmids: 0.5 μg Myc-C17orf59; 3.2 μg FLAG-CASTOR1 or FLAG-p18[14–161] or FLAG-p18[76–161]; 9.6 μg Myc-MP1 or Myc-MP1 mutants; 3.2 μg Myc-GST-p14 or Myc-GST-p14 mutants; 2.64 μg Myc-HBXIP; 1.6 μg Myc-C7orf59.

36 h after transfection, the cells were collected by centrifugation and lysed with a Triton lysis buffer (1% Triton, 10 mM β-glycerol phosphate, 10 mM pyrophosphate, 40 mM Hepes, pH 7.4, 2.5 mM MgCl₂ and EDTA-free protease inhibitor (Roche)) for 30 min at 4 °C. The cell lysates were centrifuged at 18,000 × g for 40 min, and then the supernatants were incubated with 20 μl of a 50% slurry of FLAG-M2 affinity gel (Sigma) for 2 h at 4 °C. The FLAG-M2 affinity gels were washed one time with low salt Triton wash buffer (1% Triton, 40 mM Hepes, pH 7.4, and 2.5 mM MgCl₂) and three times with high salt Triton wash buffer (1% Triton, 40 mM Hepes, pH 7.4, 500 mM NaCl, and 2.5 mM MgCl₂). Immunoprecipitated proteins were resolved by SDS-PAGE, and analyzed by Western blotting with antibodies specific to FLAG (1:3000, Sigma, F3165), Myc (1:500, Beyotime, AM926) or HA (1:3000, Sigma, H3663).

**Immunofluorescence microscopy analysis**. Immunofluorescence microscopy analysis was carried out to analyze the co-localization of the Ragulator components and the Rag GTPases. Mito or p18[14–161]-Mito (denoted as p18[mito]) or p14-Mito (p14[mito]) or C7orf59-Mito (C7orf59[mito]) was cloned into the pmRFP-C1 vector, MP1 into the pEGFP-N3 vector, RagA or RagA Gd or RagA Rd into the pEGFP-C3 vector, and Myc-MP1, Myc-p14, Myc-HBXIP, FLAG-C7orf59, Myc-C7orf59, Myc-RagC or Myc-RagC Gd, or Myc-RagC Rd into the pcDNA 3 vector.

The localization of RFP-p18[mito] was verified by Mito-Tracker Green (Beyotime). 50,000 HEK293T cells were plated on Poly-D-Lysine-coated glass coverslips in 6-well tissue culture plates, cultured in DMEM (Hyclone) supplemented with 10% fetal bovine serum (Biochrom), and then transiently transfected with the plasmids using Lipofectamine 2000 (Invitrogen). 24 h after transfection, the slides were rinsed with PBS and fixed with 4% paraformaldehyde at 25 °C for 15 min. The slides were then permeabilized with 0.05% Triton X-100 in PBS for 1 min. FLAG-C7orf59 was incubated sequentially with anti-FLAG antibody and the secondary antibody (1:1000, Sigma, AP192SA6). The coverslips were mounted on glass slides and the confocal images were obtained using a Zeiss Laser Scanning Microscope (LSM) 710 with a ×63 oil immersion lens.

For the co-localization experiments of the Ragulator components, the cells were transfected with the following plasmids: 1000 ng RFP-Mito or RFP-p18[mito] or RFP-p18[mito] mutants; 1000 ng MP1-GFP; 1000 ng Myc-p14; 1000 ng Myc-HBXIP; and 1000 ng FLAG-C7orf59. For the co-localization experiments of the Ragulator and the Rag GTPases, the cells were transfected with the following plasmids: 600 ng GFP-RagA or GFP-RagA Gd or GFP-RagA Rd; 600 ng Myc-RagC or Myc-RagC Gd or Myc-RagC Rd; 600 ng RFP-p18[14–161]-Mito or its mutants or RFP-p14[mito] or RFP-C7orf59[mito], 600 ng Myc-MP1 or its mutants, 600 ng Myc-p14 or its mutants; 600 ng Myc-HBXIP; and 600 ng Myc-C7orf59. The cells were processed for imaging as described above.

**Nucleotide exchange assay**. The GEF activity of the Ragulator complex for RagA–RagC was analyzed using the method described previously[37, 38]. Briefly, the purified RagA–RagC[D181N] was loaded with fluorescent 2′(3′)-bis-O-(N-methylanthraniloyl)-GDP (mantGDP, Invitrogen) in the presence of 10 mM EDTA and 10-fold excess of mantGDP at 4 °C overnight. Mutation D181N on RagC changes the base specificity from guanine to xanthosine nucleotides, and RagC[D181N] binds less than 2% of guanine nucleotides than wild-type protein[11]. Loading reaction was suspended by addition of 15 mM MgCl₂, and the mixture was purified by gel

filtration to remove free mantGDP in a buffer containing 20 mM Tris-HCl (pH 8.0), 150 mM NaCl, and 1 mM MgCl₂. For the GEF activity assay, 2.0 μM mantGDP-bound RagA-RagC[D181N] was pre-incubated with or without 2.0 μM Ragulator (containing p18[14–161]), and the nucleotide exchange reaction was initiated by addition of GTP to a final concentration of 10.0 μM in a final volume of 100 μl reaction buffer. Dissociation of mantGDP was monitored by measuring the decrease of fluorescence. Samples were excited at 360 nm and the emission was monitored at 440 nm. Fluorescence data were recorded using a Synergy Neo Microplate Reader (BioTek). Data were fitted against a nonlinear least-squares-fit to the exponential equation $I(t) = (I_0 - I_\infty) \exp(-k_{obs}t) + I_\infty$ where $I(t)$ is the emission intensity at time $t$, $k_{obs}$ is the observed pseudo first-order rate constant, and $I_0$ and $I_\infty$ are the emission intensities at $t = 0$ and $t = \infty$, respectively.

**Data availability**. The structures of the HBXIP-C7orf59, Ragulator (p18[49–161]), and Ragulator (p18[76–145]) complexes have been deposited in the RCSB Protein Data Bank with accession codes 5Y38, 5Y3A, and 5Y39, respectively. Other data are available from the corresponding author upon reasonable request.

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

## Acknowledgements

We thank the staff members from BL19U1 of National Facility for Protein Science in Shanghai (NFPSS) and BL17U of Shanghai Synchrotron Radiation Facility (SSRF) for technical assistance in diffraction data collection. Our microscope work was performed at the National Center for Protein Science Shanghai. This research was supported by grants from the National Natural Science Foundation of China (31530013 and 31370015) and the Chinese Academy of Sciences (XDB08010302 to JD and Youth Innovation Promotion Association of CAS to T.Z.).

## Author contributions

T.Z. carried out the structural studies. T.Z., R.W., Z.W., and X.W. carried out the functional studies. T.Z. and J.D. conceived the study, participated in the experimental design, data analyses, and discussion, and wrote the manuscript.

## Additional information

**Competing interests:** The authors declare no competing financial interests.

