## [Peer Review File · Nature Communications]

REVIEWERS' COMMENTS:

Reviewer #1 (Remarks to the Author):

The ragulator is a scaffolding complex that anchors the Rag GTPases to the lysosomal membrane. Upon activation, Rag GTPases bind mTORC1 complex which results in its activation and following regulation of cellular functions. The ragulator is a multiprotein complex containing five components. Four of them belong to the roadblock family of proteins and the fifth, p18, anchors the complex to the lysosomal membrane through the N-terminal myristoylated and palmitoylated sites in the N-terminus. The authors describe the structure of the ragulator complex and identify through mutagenesis and localization studies the binding sites for the Rag GTPases. The p18 protein plays a key role in assembling the complex and only parts that are in direct contact with the roadblock proteins are ordered. P18 ropes around the two heterodimers, p14-MP1 and C7orf59-HBXIP. The authors confirm the p18 interface observed in the crystal structure through mutagenesis and localization of all subcomponents. To avoid potential artifacts, they fuse p18 with the mitochondrial localization signal and follow the localization of the ragulator components in this unnatural location. They confirm co-localization of Rag GTPases with the ragulator and identify the N-terminal segment of p18 as essential for the co-localization. They also show that a negative regulator of mTORC1, C17orf59, binds to p14 and likely sterically interferes with Rag GTPase binding.

The manuscript is well and clearly written and, in addition to the crystal structure, presents a large body of biochemical and cellular data that support the arrangement of proteins in the ragulator complex. Moreover, the authors propose a model for the mode of action of the ragulator and in activation but also a deactivation of mTORC1.

The manuscript provides valuable information on one cellular scaffolding and localization mechanism that is general interest. I have no specific comments for the authors.

Reviewer #2 (Remarks to the Author):

The mTORC1 signaling pathway is one of the most important pathways that regulate growth in response to nutrient availability in animals, and it is implicated in many diseases. The Ragulator complex is a key complex in controlling mTORC1 activity. Therefore it is crucial to understand the molecular mechanism by which Ragulator acts on mTORC1. In this beautiful work, Zhang et al. provide compelling evidence elucidating the crystal structure of the Ragulator complex. They convincingly show that p18 wraps around the rest of the complex components and the interaction between p18 and the rest of the components is essential for the assembly of the whole complex. They identify the interaction surface between the Ragulator and the Rag GTPases. Lastly, they present evidence that the interaction between Ragulator and Rag GTPases can be inhibited by C17orf59 in a competitive manner. I have no major issues as the data are of good quality and support the conclusions drawn from them. Overall, this is a very nice study. I have only a couple minor suggestions for improvement, listed below

Minor Issues

1. In Supplementary Figure 1, where SEC-MALS data are presented, differently colored traces (green vs. blue vs. red) were not labeled. In particular, the black trace looks funny. A description of what these traces are should be included in the figure legend.
2. In Supplementary Figure 10, colocalization between Rags and mitochondria is not as clear as the other figures. Maybe the authors can find a more representative image?
3. Page 12 – the part of the sentence “whereas the p18 mutant containing the N-terminal truncation cannot,..” was difficult to understand, and I suggest rephrasing to “whereas the N-terminal truncation of p18 76-161 is capable of binding C17orf59...”

Reviewer #3 (Remarks to the Author):

Structural Basis for the Ragulator Complex Functioning as a Scaffold in Membrane-anchoring of Rag GTPases and mTORC1

This manuscript presents a clear and thorough description of the structure of the complex formed by five proteins ("Ragulator") that serve as the docking site for the Rag GTPases, and in turn mTORC1, on the lysosomal membrane. In addition, an analysis, based on mutagenesis, is presented for how the Rag proteins are expected to interact with the complex. The structure is very important as it is a key component of the fundamental process of amino acid sensing and signaling in the cell. The X-ray crystallography appears to have been carried out in a technically competent manner, despite the fact that the crystals did not diffract to very high resolution. One of the datasets is not as complete as one would typically hope for, and the Rwork/Rfree refinement statistics for this dataset are a bit high. However, the information in that structure is duplicated by the structure of a related complex variant, and certainly for the type of analysis of the complex and interactions performed in the manuscript, the quality of the structures seems sufficient. The analysis of the structures is deep, both in terms of follow-up mutagenesis and descriptions of the modes of interactions observed. Though not all information presented seems essential (e.g., Supplementary Fig. 13), the manuscript provides a comprehensive and definitive picture of the regulator complex and its structural relationship to other complexes involving proteins with longin/roadblock domains.

Some very minor corrections are recommended:

Line 90: should be "could be co-expressed"

Lines 91-92: Perhaps replace "a stable pentameric complex which exists as a monomer in solution as revealed by" with "a stable, monodisperse, hetero-pentameric complex as revealed by"

Line 97: should be "residues 1-48 of p18 are not required"

Line 98: should be "are necessary and sufficient"

Line 107: Coiled coil? It does not seem that p18 forms a coiled coil in the conventional sense used in structural biology: two helices wrapped around one another with a superhelical twist.

Line 189: should be "Despite lacking"

There is a missing period in line 67 in the main text and an extra one in line 37 in the supplementary.

Responses to comments by the reviewers

Reviewer #1

Overall comments: The ragulator is a scaffolding complex that anchors the Rag GTPases to the lysosomal membrane. Upon activation, Rag GTPases bind mTORC1 complex which results in its activation and following regulation of cellular functions. The ragulator is a multiprotein complex containing five components. Four of them belong to the roadblock family of proteins and the fifth, p18, anchors the complex to the lysosomal membrane through the N-terminal myristoylated and palmitoylated sites in the N-terminus. The authors describe the structure of the ragulator complex and identify through mutagenesis and localization studies the binding sites for the Rag GTPases. The p18 protein plays a key role in assembling the complex and only parts that are in direct contact with the roadblock proteins are ordered. P18 ropes around the two heterodimers, p14-MP1 and C7orf59-HBXIP. The authors confirm the p18 interface observed in the crystal structure through mutagenesis and localization of all subcomponents. To avoid potential artifacts, they fuse p18 with the mitochondrial localization signal and follow the localization of the ragulator components in this unnatural location. They confirm co-localization of Rag GTPases with the ragulator and identify the N-terminal segment of p18 as essential for the co-localization. They also show that a negative regulator of mTORC1, C17orf59, binds to p14 and likely sterically interferes with Rag GTPase binding.

The manuscript is well and clearly written and, in addition to the crystal structure, presents a large body of biochemical and cellular data that support the arrangement of proteins in the ragulator complex. Moreover, the authors propose a model for the mode of action of the ragulator and in activation but also a deactivation of mTORC1.

The manuscript provides valuable information on one cellular scaffolding and localization mechanism that is general interest. I have no specific comments for the authors.

Response: We appreciate the positive comments by this reviewer.

Reviewer #2:

Overall comments: The mTORC1 signaling pathway is one of the most important pathways that regulate growth in response to nutrient availability in animals, and it is implicated in many diseases. The Ragulator complex is a key complex in controlling mTORC1 activity. Therefore it is crucial to understand the molecular mechanism by which Ragulator acts on mTORC1. In this beautiful work, Zhang et al. provide compelling evidence elucidating the crystal structure of the Ragulator complex. They convincingly show that p18 wraps around the rest of the complex components and the interaction between p18 and the rest of the components is essential for the assembly of the whole complex. They identify the interaction surface between the Ragulator and the Rag GTPases. Lastly, they present evidence that the interaction between

Ragulator and Rag GTPases can be inhibited by C17orf59 in a competitive manner. I have no major issues as the data are of good quality and support the conclusions drawn from them. Overall, this is a very nice study. I have only a couple minor suggestions for improvement, listed below

Response: We appreciate the positive comments by this reviewer.

Minor comments

Comment 1: In Supplementary Figure 1, where SEC-MALS data are presented, differently colored traces (green vs. blue vs. red) were not labeled. In particular, the black trace looks funny. A description of what these traces are should be included in the figure legend.

Response: We have added a detailed description for these traces in the figure legend of Supplementary Figure 1 as follows: “The left and right vertical axes represent the refractive index reading and the molecular mass. Chromatograms show the readings from the light scattering at 90° (red), refractive index (blue), and UV (green) detectors. The black curves represent the calculated molecular masses, and the average measured masses of the elution peaks of the Ragulator complex (p18¹⁴⁻¹⁶¹) and the C7orf59-HBXIP heterodimer are indicated, suggesting that both complexes exist as monomers in solution.”

Comment 2: In Supplementary Figure 10, colocalization between Rags and mitochondria is not as clear as the other figures. Maybe the authors can find a more representative image?

Response: We thank the reviewer for the suggestion. In Supplementary Figure 10, RagA^{GTP}-RagC^{GDP} and RagA^{GDP}-RagC^{GTP} colocalized with p18^{mito} compared with the negative results in Supplementary Figure 12. The colocalization of RagA^{GDP}-RagC^{GTP} is clearer than that of RagA^{GTP}-RagC^{GDP}. We noticed this difference during preparation of this manuscript, and thus repeated these experiments several times but obtained similar results. We speculate that it might be due to the lower binding affinity between the Ragulator complex and RagA^{GTP}-RagC^{GDP}. Indeed, our co-IP experiments show that the Ragulator complex can interact with both RagA^{GTP}-RagC^{GDP} and RagA^{GDP}-RagC^{GTP} but it pulls down more RagA^{GDP} than RagA^{GTP} (Supplementary Figure 10a). The colocalization and co-IP assay results are in agreement with each other. This observed difference has been briefly discussed in the Discussion (Lines 290-292).

Comment 3: Page 12 – the part of the sentence “whereas the p18 mutant containing the N-terminal truncation cannot,..” was difficult to understand, and I suggest rephrasing to “whereas the N-terminal truncation of p18 76-161 is capable of binding C17orf59...”

Response: We thank the reviewer for the suggestion. This sentence has been rephrased as: “whereas the N-terminal truncation of p18 (residues 76-161) is capable of binding C17orf59”.

Reviewer #3:

Overall comments:

This manuscript presents a clear and through description of the structure of the complex formed by five proteins (“Ragulator”) that serve as the docking site for the Rag GTPases, and in turn mTORC1, on the lysosomal membrane. In addition, an analysis, based on mutagenesis, is presented for how the Rag proteins are expected to interact with the complex. The structure is very important as it is a key component of the fundamental process of amino acid sensing and signaling in the cell. The X-ray crystallography appears to have been carried out in a technically competent manner, despite the fact that the crystals did not diffract to very high resolution. One of the datasets is not as complete as one would typically hope for, and the Rwork/Rfree refinement statistics for this dataset are a bit high. However, the information in that structure is duplicated by the structure of a related complex variant, and certainly for the type of analysis of the complex and interactions performed

In the manuscript, the quality of the structures seems sufficient. The analysis of the structures is deep, both in terms of follow-up mutagenesis and descriptions of the modes of interactions observed. Though not all information presented seems essential (e.g., Supplementary Fig. 13), the manuscript provides a comprehensive and definitive picture of the regulator complex and its structural relationship to other complexes involving proteins with longin/roadblock domains.

Response: We appreciate the positive comments by this reviewer.

Minor comments

Comment 1: Line 90: should be “could be co-expressed”.

Response: We thank the reviewer for pointing out this error, which has been corrected in the revision.

Comment 2: Lines 91-92: Perhaps replace “a stable pentameric complex which exists as a monomer in solution as revealed by” with “a stable, monodisperse, hetero-pentameric complex as revealed by”.

Response: We thank the reviewer for the suggestion. This sentence has been rephrased as suggested.

Comment 3: Line 97: should be “residues 1-48 of p18 are not required”

Response: We thank the reviewer for pointing out this error, which has been corrected in the revision.

Comment 4: Line 98: should be “are necessary and sufficient”

Response: We thank the reviewer for pointing out this error, which has been corrected in the revision.

Comment 5: Line 107: Coiled coil? It does not seem that p18 forms a coiled coil in the conventional sense used in structural biology: two helices wrapped around one another with a superhelical twist.

Response: We concur with the reviewer that the use of “coiled coil” here is not very precise. In the revision, we have rephrased the sentence as follows: “p18 forms a helical structure...” (Line 108).

Comment 6: Line 189: should be “Despite lacking”

Response: We thank the reviewer for pointing out this error, which has been corrected in the revision.

Comment 7: There is a missing period in Line67 in the main text and an extra one in Line 37 in the supplementary.

Response: We thank the reviewer for pointing out these errors, which have been corrected in the revision.